# Advances in Innate Immunity to Overcome Immune Rejection during Xenotransplantation

**DOI:** 10.3390/cells11233865

**Published:** 2022-11-30

**Authors:** Tian-Yu Lu, Xue-Ling Xu, Xu-Guang Du, Jin-Hua Wei, Jia-Nan Yu, Shou-Long Deng, Chuan Qin

**Affiliations:** 1NHC Key Laboratory of Human Disease Comparative Medicine, Institute of Laboratory Animal Sciences, Chinese Academy of Medical Sciences and Comparative Medicine Center, Peking Union Medical College, National Human Diseases Animal Model Resource Center, Beijing Engineering Research Center for Experimental Animal Models of Human Critical Diseases, International Center for Technology and Innovation of animal model, Beijing 100021, China; 2National Engineering Laboratory of Animal Breeding, College of Animal Science and Technology, China Agricultural University, Beijing 100193, China; 3State Key Laboratory of Agrobiotechnology, College of Biological Sciences, China Agricultural University, Beijing 100193, China; 4Cardiovascular Surgery Department, Center of Laboratory Medicine, National Center for Cardiovascular Diseases, Fuwai Hospital, Chinese Academy of Medical Sciences and Peking Union Medical College, Beijing 100037, China; 5Changping National Laboratory (CPNL), Beijing 102206, China

**Keywords:** xenotransplantation, innate immune system, immunological rejection, complement, innate immune cell, TLR

## Abstract

Transplantation is an effective approach for treating end-stage organ failure. There has been a long-standing interest in xenotransplantation as a means of increasing the number of available organs. In the past decade, there has been tremendous progress in xenotransplantation accelerated by the development of rapid gene-editing tools and immunosuppressive therapy. Recently, the heart and kidney from pigs were transplanted into the recipients, which suggests that xenotransplantation has entered a new era. The genetic discrepancy and molecular incompatibility between pigs and primates results in barriers to xenotransplantation. An increasing body of evidence suggests that innate immune responses play an important role in all aspects of the xenogeneic rejection. Simultaneously, the role of important cellular components like macrophages, natural killer (NK) cells, and neutrophils, suggests that the innate immune response in the xenogeneic rejection should not be underestimated. Here, we summarize the current knowledge about the innate immune system in xenotransplantation and highlight the key issues for future investigations. A better understanding of the innate immune responses in xenotransplantation may help to control the xenograft rejection and design optimal combination therapies.

## 1. Introduction

The inadequate supply of human organs for allografting has prompted interest in xenografts. Xenotransplantation is one of the most attractive strategies for alleviating the critical worldwide shortage of donor organs for clinical transplantation. Therefore, pig-to-human xenotransplantation has become a promising field of research to overcome this problem. There are three main types of immune rejection after xenotransplantation. Hyperacute rejection (HAR) occurs within minutes or hours. An acute humoral xenograft rejection (AHXR) evolves within days or weeks, causing organ xenograft dysfunction. Additionally, an acute cellular rejection is another common type of allogeneic rejection [1]. In addition to immune rejection, coagulation dysfunction and inflammatory reactions have been observed, leading to xenograft failure. The role of innate immunity as a significant driver of immunological rejection of xenografts is increasingly recognized.

The first major obstacle to xenotransplantation is the hyperacute rejection of the transplanted organ due to natural antibodies. After that, cellular immunity, including T and B lymphocytes, natural killer (NK) cells, and macrophages may become involved as well. Currently, because of the strong immune response to xenografts, success will probably depend upon the new strategies for immunosuppression.

Histopathological studies have demonstrated that the mechanisms involved in xenogeneic graft rejection are significantly different from those associated with allogeneic graft rejection. Cytotoxic T lymphocytes may play a major role in allograft rejection, whereas xenografts mainly induce the infiltration of NK cells and macrophages [2,3]. Consequently, innate immune responses play a more prominent role in the rejection of xenografts than in allograft rejection.

Innate immunity is an important system in animals and humans that is precisely regulated and used to maintain homeostasis and prevent a microbe invasion. Innate immunity is coordinated by immune cells, natural antibodies, and complements, along with cytokines and chemokines secreted by innate immune cells under physiological and pathological conditions [4,5]. The activity of innate immune cells mainly comprises macrophages, neutrophils, macrophages, NK cells, and several other cell types that dominate innate immune responses [6,7,8]. The cells of the innate immune system are evolved to express the pattern recognition receptors (PRRs) that recognize specific pathogen-associated molecular patterns (PAMPs), and mount a specific immune response [9]. Toll-like receptors (TLRs) are one of these PRRs which are expressed by various immune cells [10]. These innate immune cells not only participate in the occurrence of injury and inflammation, but also in the timely clearing of damaged cells including apoptotic and necrotic cells. Alternatively, they induce the maturation of antigen-presenting cells (APC) and the migration to secondary lymphoid tissues where they trigger primary T cell and B cell immunity. Disclosing the sophisticated actions of innate immunity will facilitate the strategies of limiting inflammation, promoting repair, and extending xenograft survival.

The reviews published in the past few years mainly discuss the functions of the adaptive immune system and the initiation of the allo-immune response in general. The current armamentarium of immunosuppressive drugs is primarily designed to limit the adaptive immune response. The role of innate immunity as a significant driver of the xeno-immune response is increasingly recognized. However, only a few foci were placed on the immune effects and immune treatment strategies of the innate immune system after xenotransplantation. In this review, we summarize recent advances in the role of the innate immune system in xenotransplantation, as well as the innate immune system-based immunotherapies strategies to overcome xenogeneic rejection.

## 2. The Complement and Natural Antibodies System

As we know, the complement system still plays a vital role in xenogeneic rejection [11]. Complements of humans and other mammals may be activated in a xenograft (Table 1). The complement system is activated through three distinct pathways: the classical pathway (CP), driven by antibody binding and Fc-mediated complement activation; the lectin pathway (LP); and the alternative pathway (AP), triggered by spontaneous unknown mechanisms [12,13]. All three of these pathways generate C3 convertases that cleave C3 to C3a and C3b. In turn, C3b and C5b lead to the formation of the C5b-9 ‘membrane attack complex’ (MAC), which results in the activation of granulocytes, endothelia, and epithelia or cell death [12] (Figure 1). C5a and C3a are chemoattractants for neutrophils and macrophages that act on specific receptors to produce local inflammatory responses [14]. In addition to being a central part of complement activation, complement C3 is a functional bridge between innate and adaptive immune responses [15,16].

The activation of a complement is controlled in part by complement regulatory proteins (CRPs) that are the integral components of cell membranes. EC activation in the xenograft setting is a key step in the pathophysiology of AVR, and the data suggest that the classic complement pathway plays a major role in the human serum-induced activation of porcine endothelial cells (ECs) [17]. The first breakthrough in xenotransplantation research was the report of species differences in the complement system, which found that the rejection in guinea pigs of rat xenotransplantation models was caused by the primary activation of the complement via a surrogate pathway [18].

The inhibition activation of the complement system can prevent the occurrence of HAR in pig to non-human primate (NHP) xenotransplantation. The complement regulatory proteins play a role in a species-specific manner to discriminate between the self and non-self to prevent the over-activation of the complement system and complement-mediated damage. These proteins include the decay-accelerating activator (DAF, also known as CD55), the membrane inhibitor of reactive lysis (CD59), and the membrane cofactor protein (MCP, also known as CD46). CD46 is a cofactor for the factor I-mediated inactivation of C3b and C4b [19]. Human complement regulatory protein (hCRP) transgenic pigs were created to enhance the protection of endothelial cells (pECs) against human complement-mediated injury. Baboons were xenografted with multiple-transgenic pig livers expressing the hCD55, hCD59, and human α1,2-fucosyltransferase (H-transferase, HT) to prevent HAR and allow for the initiation of the liver function, including the synthesis of coagulation factors and progressively increased prothrombin activity [20]. However, there is evidence that the expression of the hCD55 eliminated an early graft failure and restricted the complement activation in cardiac xenografts transplanted into baboons but did not improve the graft survival [21,22].

Chronic rejection is also a potential cause of clinical allograft failure. Indeed, xenograft vasculopathy has been reported in the hearts transplanted from hCRP transgenic pigs to primates, which was considered to be a characteristic feature of chronic rejection [23,24]. The main manifestations are the proliferation of endothelial cells in the graft vessels, vessel narrowing, and thrombosis, which eventually leads to graft failure [25]. Fortunately, the use of anti-CD40 mAb on transgenic pig hearts (GTKOhTg.hCD46.hTBM) blocked the same CD40-CD40 ligand costimulation pathway to avoid thrombotic microangiopathy and allow for the successful survival of pig cardiac xenografts in baboons for 236 days [26]. The result suggested that anti-CD40 mAb can achieve a long-term cardiac xenograft survival. Additionally, the expression of hCD39 on mouse hearts attenuated a platelet deposition and small vessel thrombosis, thereby prolonging the graft survival, and the cardiac xenografts from CD39-null mice in rats showed reduced survival times [27]. CD4 + T cells may be a cause of chronic rejection. The depletion of CD4 T cells maintained the survival of pig to rhesus macaque kidney xenografts for up to 499 days [28].

Studies have shown that dextran sulfate (DXS), which inhibits the classical and alternative complement pathway, can delay hyperacute rejection in xenotransplantation models [29,30]. Similarly, Laumonier et al. have reported that DXS dose-dependently inhibits all three pathways of the complement activation and the preferential binding to activated ECs that have lost their protective layer of heparan sulfate proteoglycan (HSPG), and thus protects the endothelium from a human complement-mediated injury [31].

Natural antibodies (Nab) are produced early in life and constitute life-long high titers of circulating antibodies. Most complement-fixing xenoreactive Nab in humans is directed against αGal, a saccharide expressed in the cells of lower mammals [32]. The αGal epitope does not exist in human and Old-World monkeys, as they lack the functional alpha1,3-galactosyltransferase enzyme (α1,3GalT; also known as GGTA1). Human and Old-World monkeys have high titers of anti-αGal antibodies due to an exposure to αGal-expressing bacteria in the gastrointestinal tract [33]. Almost as soon as the xenograft is perfused, Nabs bind to αGal on pECs, and causes the HAR of the graft through the engagement of the classical complement pathway and the coagulation system.

In pig-to-primate organ transplantation, HAR can also be prevented by depleting the recipient of the anti-αGal or anti-non-Gal antibody [34]. Several groups have reported that the use of these transgenic pig organs in the NHP can prevent HAR and can extend the xenograft survival. Compared with wild-type pig donors, the use of hCD46/GTKO gene-modified pig islets can significantly improve the survival time of the xenografts [35]. With the development of GTKO pigs, donor organs became available in which the expression of αGal was eliminated [36]. A heart or kidney transplantation in baboons using GTKO pigs as donors effectively prevents HAR [37,38]. Therefore, the complete removal of the αGal epitope from pig organs can eliminate the major carbohydrate antigen recognized by the human anti-pig antibody [39].

The addition of multiple hCRPs and HT to GTKO further reduced the incidence of initial graft dysfunction by constraining the amplification of the antibody-mediated classical complement cascade, inhibiting the formation of the MAC, and better preventing an endothelial injury relative to either hCPRP or GTKO alone [20,40]. In addition, the data from GTKO rodents suggested that isoglobotriaosylceramide-3 synthase (iGb3s), is an alternative galactosyltransferase that can attenuate the expression of αGal [41]. However, although iGB3s mRNA can be detected in selected porcine tissues, residual αGal expression has not been recorded in the pECs derived from GTKO pigs to date [42,43]. In addition, studies suggest that the removal of non-Gal antibodies in pigs also reduces the binding of IgG/IgM to xenograft [44]. Therefore, additionally modified CMP-N-acetylneuraminic acid hydroxylase (CMAH) and beta-1,4-N-acetyl-galactosaminyltransferase 2 (β4GALNT2) gene knockout pigs were produced [45,46,47]. Martens et al. revealed that swine leukocyte antigens (SLA) class I is crucial to the cellular immune response in pig-to-human xenotransplantation by screening for human antibody binding using GGTA1/CMAH/B4GalNT2 and SLA class I gene KO pig cells [48]. HLA is a protein complex expressed on human tissue that can induce a hyperacute, acute, or chronic rejection in a transplantation, leading to graft failure. The knockout or knockdown of the genes encoding the SLA molecule can ensure the long-term survival of xenografts. Non-human primates (NHPs) typically carry naturally occurring pig cell-specific preformed antibodies that are not always present in human serum [49]. The antibody binding to Neu5Gc-deficient pigs was also increased in Old-World monkeys, which activated the CMAH gene and have natural antibodies against Neu5Gc-deficient pigs [50]. However, the porcine cells lacking the three major carbohydrate xenoantigens, αGal, Neu5GC, and SDa demonstrated a significantly reduced Nab binding and transplanted the renal xenografts from triple-knockout pigs expressing multiple human transgenes in cynomolgus monkeys, resulting in achieving a long-term, non-rejection xenograft survival [51]. Therefore, we hold that preliminary studies in pigs lacking the three identified xenoantigens with the expression of one or more hCRPs have shown promise in xenografts and will be greatly developed in the future.

At the same time, the study has shown that by supplementing the primate blood coagulation components, massive hemorrhage in the setting of a liver xenotransplantation can be avoided [52]. In 2017, the team from Massachusetts General Hospital extended the survival time of a liver xenotransplantation from 8 days to 29 days. It depends on the use of a GTKO pig donor and the administration of the CD40 antibody, belatacept, and postoperative coagulation factor complex [53]. Furthermore, as a preventive method targeting the complement pathway, pigs expressing transgenes for hCRPs, and miniature pigs with quadruple modified genes cytidine monophosphate-N-acetylneuraminic acid hydroxylase (CMAH)KO/GTKO/soluble human tumor necrosis factor receptor I IgG1-Fc (shTNFRI-Fc)/human hemagglutinin-tagged-human heme oxygenase-1 (hHO-1) were developed [54,55].

## 3. The Innate Cellular Immune Response in Xenotransplantation

### 3.1. Macrophages-Mediated Xenograft Rejection

The primary interaction of macrophages with human cells appears to be mediated by lectins expressed by the human macrophages, which are stimulated by endothelial cell saccharides (Table 1). Once stimulated by this or other interactions, the macrophages may initiate coagulation through the elaboration of the tissue factors.

The first indications of the involvement of macrophages in xenorejection came from studies demonstrating that the cellular infiltrates in rejected xenogeneic heart grafts in hamster- or guinea-pig-into-rat or pig-into-baboon models contained a prominent macrophage population [2,56,57]. After xenotransplantation, the macrophages are activated and quickly recruited, and their response to xenoantigens occurs before adaptive immune activation [58].

The macrophages activated by T cells were attracted from the peripheral circulation and were capable of the specific targeting and destruction of porcine islet grafts [59]. O’Connell et al. have shown in mice that the high expression of MCP-1, macrophage inflammatory protein (MIP)-1alpha, MIP-1beta, and being regulated upon activation may be important in the recruitment of macrophages to porcine islet grafts, and that the expression of TLRs, chemokine (C-C motif) receptor 2 (CCR2), and the chemokine (C-C motif) receptor 5 (CCR5) is upregulated in activated macrophages [60]. Similarly, Wu et al. found that mouse macrophages were depleted by an intraperitoneal injection of liposome-encapsulated dichloroethylene diphosphonate (Lip-Cl2MDP) which may facilitate pig-into-mouse xenogeneic islet survival [61]. Andres et al. treated streptozotocin-induced diabetic C57BL/6 mice with gadolinium chloride to deplete the macrophages, and transplanted human islets under the kidney capsule showed that macrophage depletion allowed for the prolongation of an islet xenograft survival by analyzing the histology and immunohistochemistry of the xenograft [62]. The treatment of baboons with medronate liposomes to deplete macrophages has been shown to greatly increase the level and duration of the xenogeneic chimerism, but this treatment is toxic and incompatible with tolerance-inducing mechanisms that rely on the co-stimulation of the blockade [63,64].

The pathways by which macrophages recognize and destroy xenogeneic tissues are gradually being discovered. Macrophages can be activated through Fc receptors (FcRs) and complement receptors, and hence participate in a complement activation and the xenoreactive natural antibody-dependent mechanisms of xenograft rejection (Figure 2). Signal-regulatory protein-α (SIRPα) is abundantly expressed on polymorphonuclear leukocytes, monocytes, and monocyte-derived cells, where it is an inhibitory receptor. CD47, a phagocytosis checkpoint in macrophages, binds to SIRPα and inhibits the clearance of cells by the immune system [65]. Evidence suggests that macrophages can directly recognize xenogeneic cells through the SIRPα-CD47 pathway, thereby playing a prominent role in signalling by preventing phagocytosis [66].

In addition, studies suggested that the decreased clearance of porcine cells in primates may be achievable by the transgenic expression of primate CD47. For example, the transgenic expression of human CD47 (hCD47) on porcine cells substantially increased the duration of the transient xenogeneic chimerism following the transplantation of hematopoietic cells (HCs) to baboons [67]. Similarly, the non-obese diabetic (NOD)–severe combined immunodeficiency (SCID) mouse SIRPα is known to be capable of interacting with hCD47 [68]. SIRP-α showed an enhanced binding to the hCD47 ligand in NOD/SCID mice, influencing the survival and engraftment of human hematopoietic stem cells (HSCS) [68]. The genetic induction of hCD47 on porcine cells reduces the susceptibility of porcine cells to phagocytosis by human macrophages in vitro [69,70]. This is due to SIRPα, a key immune inhibitory receptor on macrophages, which effectively binds hCD47 to prevent autologous phagocytosis. Moreover, Tena et al. reported that hematopoietic progenitor cells from transgenic pigs expressing hCD47 reduced the macrophage-mediated phagocytosis of xenogeneic target cells in a murine model of human bone marrow engraftment [71]. It has been suggested that transgenic pigs expressing hCD47 as donors may be a potential pathway for clinical xenotransplantation.

By contrast, thrombospondin 1 (TSP-1) is an adhesive glycoprotein that mediates cell behavior by engaging with molecules in the extracellular matrix and with receptors on the cell surface and contributes to platelet aggregation. CD47 is a high-affinity TSP1 receptor. TSP1-CD47 signaling has been indicated to accelerate the rejection of vascularized allografts [72]. TSP-1 also blocks hCD47-SIRPα signalling and increases macrophage phagocytosis. However, Watanabe et al. suggested that hCD47 expression in both alveoli and vessels may play a beneficial role in prolonging lung xenograft survival [73]. Further investigations of the strong and weak points of CD47 in xenografts are needed because the function of the macrophages varies by organ.

CD200 is another inhibitory molecule involved in the regulation of macrophage proinflammatory immune responses. CD200 induces an immunosuppressive function through the binding to CD200R, which contains an inhibitory intracellular Asn-Pro-X-Tyr (NPXY) signaling motif. Human CD200 on porcine cells was found to suppress macrophage-mediated phagocytosis and cytotoxicity. Yan et al. demonstrated that the overexpression of CD200 in pECs suppressed the xenogeneic immune response of human macrophages and improved the survival of pEC xenografts in humanized mice [74]. Expressing CD200 could represent a novel approach to prolonging xenograft survival.

The surfactant protein (SP)-D molecules recognize PAMPs on microorganisms to induce innate immunity and contribute to the pathogen removal and pro-inflammatory responses to infection. SP-D can also bind SIRPα, decreasing the cytokine production by monocytes/macrophages. Jiaravuthisan et al. prepared cDNA for the membrane-type protein, collectin placenta 1 (CL-P1) with the C-terminal heads of SPD, and transfected it into EC, then found that this molecule was able to suppress macrophage-mediated xenobiotic cytotoxicity [75]. In addition, in vitro studies have shown that Cl-SPD has a stronger inhibitory effect on the macrophage-mediated cytotoxicity than CD47.

CD33-related sialic acid-binding immunoglobulin-like lectins (Siglecs) belongs to the immunoglobulin superfamily that typically exerts their functions by binding to sialylated ligands [76,77]. Human macrophages express the various CD33-related Siglecs. Therefore, the transgenic expression of α-2,6-sialyl transferase (α 2,6-ST) induces the binding of the α-2,6-sialylated ligands to CD33-related Siglecs on the macrophages, thereby inhibiting macrophage-mediated xenocytotoxicity in vitro [78]. The findings suggested that α 2,6-ST transgenic expression may contribute to macrophage-mediated xenocytotoxicity [78]. However, the in vivo function of α 2,6-ST is unclear, and further investigations on the function of α 2,6-ST in animals are needed.

### 3.2. Natural Killer Cells-Mediated Xenograft Rejection

Evidence indicates that the mechanism of the cytotoxicity of NK cells to xenogeneic cells has at least three separate pathways (Table 1). First, the molecular incompatibility between the major histocompatibility complex (MHC) class I on the xenogeneic target and killer inhibitory receptors of the natural killer cells result in NK cell activation and direct perforin-mediated damage or cytokine production that augments specific T cell immunity [79]. Second, binding to xenogeneic cells by stimulating the Fcγ II receptors xenoreactive IgG on NK cells. Finally, the stimulation of the lectin receptors on NK cells by saccharides such as Gal on the xenograft then induces antibody-dependent cell cytotoxicity (ADCC). Although most attention has focused on the possibility that NK cells may attack and destroy xenografts, recent studies suggest that NK cells also promote antigen presentation, thereby eliciting the immune responses directed against the xenograft. Alternatively, human NK cells may also be involved in innate immune responses to xenografts through the interaction of FcγRIII (CD16) with anti-αGal antibodies [80].

NK cells can destroy xenogeneic tissues through direct lysis and ADCC. AHXR is characterized by multiple microvascular thrombosis and massive NK-cell and macrophage infiltration. Additionally, the results of the study of Chen et al. indicated that in the absence of intravascular thrombosis, xenograft rejection in the mouse-to-rat heart transplantation model is mediated by NK cells [81]. In addition, Khalfoun et al. developed an experimental approach using pig kidneys perfused with human peripheral blood lymphocytes which demonstrated the predominance of a xenograft infiltration by NK cells [82].

NK cells are tightly regulated through the signals mediated by inhibiting and activating receptors expressed on their cell surface. Many inhibitory NK receptors recognize MHC class I molecules, and in the absence of sufficient inhibitory signals, the binding of activating receptors induces NK cytotoxicity [83] (Figure 2). The in vitro susceptibility of pECs to human NK cytotoxicity may be the result of the lack of recognition and inability of SLA class I to signal through human NK inhibitory receptors [84]. Second, NK cells contribute to a xenograft injury through mechanisms other than direct cytotoxic lysis. The expression of CD16 enables human NK cells to mediate ADCC. NK cells induce an endothelial cell activation, promoting the adhesion and migration of immune cells, which results in changing the nature of the endothelial cell’s surface from an anticoagulant surface to a procoagulant surface and enhances cytokine secretion, such as the tumor necrosis factor (TNF)-α and interferon (IFN)-γ [80,85]. Yin et al. demonstrated in a rodent model that NK cells mediate the IgG1-dependent hyperacute rejection of xenografts [86]. In addition, based on the results of pig-to-primate animal models, NAb binding to αGal on pECs induced a complement activation, lysis, and hyperacute rejection in pig-to-human xenotransplantation [87].

In addition to a direct NK cell interaction with the vascular endothelium leading to an endothelial injury, NK cells can migrate to the graft, providing cross-species interactions between NK cells, and pEC-related adhesion molecules can occur. Here, it is worth noting that the lack of Gal on the ECs on GalT-KO pigs resulted in a significant reduction in complement-mediated lysis and ADCC, but did not eliminate NK cells adhesion and direct anti-porcine NK cytotoxicity, indicating that αGal is not a major target for direct human NK cytotoxicity against porcine cells [88]. There is the binding of non-Gal xenoreactive antibodies to porcine ECs and direct NK-cell lysis in xenotransplantation should not be underestimated.

Analyzing adhesive interactions between purified hNK cells and pECs under static conditions by the blocking of common adhesion receptors demonstrated the involvement of CD2, CD11a (LFA-1 integrin-αL chain), CD11b (Mac-1 integrin-αM), CD18 (integrin-β2), and CD49d (VLA-4 integrin-α4) on hNK cells and CD106 on pEC [89]. After blocking the anti-porcine-CD106 monoclonal antibody (mAb) can inhibit the hNK cell adhesion to pEC to 95% [89]. Thus, blocking CD106 is an attractive target for protecting porcine xenografts from hNK cytotoxicity.

The activating receptor NKG2D has been found to be capable of mediating human anti-pig NK cytotoxicity. ULBP1 is the primary, functional porcine ligand for human NKG2D, which plays a critical role in the cytotoxic lysis of pECs by hNK cells through the recognition of ligands ULBP1. Lilienfeld et al. indicated that using anti-ULBP1 polyclonal completely blocked hNK cytotoxic lysis against pEC [90].

Further, an investigation tested a range of NK receptors, including NKp46, 2B4, CD49d, CD48, CD2, and NKG2D, and found that only CD2 and NKG2D were involved in the cytotoxicity and cytokine (IFN-γ and TNF-α) release against porcine cells [91]. It further provides an effective method to suppress xenogeneic NK responses in a porcine-human transplantation model by blocking CD2 and NKG2D. In addition, extracellular signal-regulated kinase (ERK) plays an important role in NK-mediated xenoreactivity, and the addition of PD98059, an ERK kinase inhibitor, reduces the NK cytotoxicity. Indeed, considering that NK cells consist of a heterogeneous population of cells expressing different sets of activated and inhibited receptors, it may be necessary to simultaneous target multiple receptors.

HLA-E and HLA-G are immunoregulatory molecules that belong to the HLA-Ib family. The main function of HLA-Class Ib molecules is to regulate the immune response by interacting with the specific inhibitory receptors expressed on different immune effector cells [92]. Previous experiments demonstrated that the transgenic expression of HLA-E and G suppresses NK-mediated cytotoxicity. The vast majority of macrophages express inhibitory receptors, including the CD94/NKG2A heterodimer and ILT-2,4 which recognize human HLA-E and HLA-G1 [93]. Transgenic HLA-E pigs notably inhibited macrophage-mediated xenogeneic cytotoxicity and proinflammatory cytokine production, and the results also showed that the HLA-E transgenic expression was suppressed to an extent comparable to the CD47 transgene expression [93]. Another study suggested that the production of HLA-E and G1 transgenic pigs reduced NK cell- and macrophage-mediated cytotoxicity [94].

CD154, an inflammatory mediator also known as CD40 ligand, has been identified as a novel binding partner for some members of the integrin family. Interleukin (IL)-2-activated human NK cells can express CD154 (CD40 ligand), which is an inflammatory mediator of the TNF family. NK cells also interact with marginal zone B cells in the spleen via the CD40-CD154 pathway and the production of T cell-mediated αGal-independent antibodies, thereby facilitating antibody-mediated rejection [95]. Therefore, anti-CD154 mAb can suppress an NK-mediated xenogeneic rejection. An anti-CD154 mAb, a powerful co-stimulation blockade agent, has been reported to suppress rejection, even in a xenogeneic environment [96]. In diabetic monkeys with pig islet grafts, anti-CD154 mAb prevented the rejection of genetically engineered pig islets in monkeys.

It has been shown that prenylated quinolinecarboxylic acid-18 (PQA-18), a novel immunosuppressant, could suppress the production of IL-2, IL-4, IL-6, and TNF-α production in human peripheral lymphocytes [97]. In addition, PQA-18 greatly reduced the expression of HLA-DR, CD11b, and CD40 in macrophages as well [98]. Therefore, based on these findings, the combination of anti-cd154 mAb and PQA-18 treatment is expected to produce the greater inhibition of a xenogeneic rejection.

Taken together, the implementation of strategies to inhibit NK cytotoxicity and other yet to be defined NK cell responses, such as the cytokine release and enhancement of adaptive immune responses, may promote both a successful clinical xenotransplantation. Paradoxically, despite the above studies elucidating a role for NK cells in rejection, in some animal models, they are also involved in immune regulation and are necessary for allograft tolerance [84]. Specifically, among the potential mechanisms by which NK cells can contribute to a transplant tolerance induction are the regulation of T cell responses and the antigen-presenting cell function.

### 3.3. Neutrophils-Mediated Xenograft Rejection

Neutrophils are generally viewed as nonspecialized effector cells that are recruited into tissues as a response to inflammatory mediators and through the formation of C3bi on foreign cell surfaces. Indeed, there may also exist primary interactions between neutrophils and xenogeneic cells that might precede inflammation. For example, human neutrophils attach to pECs and activate these cells independently of xenoreactive antibodies, complement, or other plasma mediators (Table 1). Neutrophils can also modulate graft inflammation by undergoing a unique form of programmed cell death termed “NETosis” [99].

Activated neutrophils can induce xenograft damage primarily by the production of reactive oxygen species (ROS), the release of tissue digestive enzymes, or neutrophil extracellular traps (Figure 2).

The complement is thought to increase the adhesion of neutrophils [100]. The incubation of decellularized porcine tissue with human plasma resulted in sequential IgG adsorption, and the activation of the classical complement pathway with a prominent adhesion of neutrophils was observed [101]. A potent complement C3 inhibitor, compstatin analogue Cp40, has been elucidated to eliminate the adhesion of leukocytes and, more specifically, neutrophil adhesion to xenogeneic target cells [102].

The activation of human neutrophils by pECs is independent of αGal structures [103]. Human neutrophils can directly recognize xenogeneic endothelium, resulting in increased synthesis and the expression of vascular cell adhesion molecule-1 on the ECs, and the enhanced killing of the xenogeneic endothelium by NK cells [104]. The binding of neutrophils to xenografts depends on the interaction between CD11a and CD11b/CD18 (MAC-1) and their ligands for CD11a and CD11b/CD18 suppress neutrophil adhesion in vitro [104]. This adhesive interaction was found to equally activate the neutrophils, with the enhanced production of the reactive oxygen metabolite and upregulating some proinflammatory cytokines, such as IL-1α, IL-6, IL-1β, and IL-8 [103] (Figure 2).

The cluster of differentiation 31 (CD31), also known as the platelet EC adhesion molecule 1, is a member of the Ig-ITIM family and is expressed on hemopoietic and ECs where it functions as a homophilic adhesion and signaling receptor [105]. Recent studies have shown that porcine cells overexpressing human CD31 suppress neutrophil NETosis and cytotoxicity in a CD31 homophilic ligation-dependent manner [106]. Furthermore, CD31 is incompatible between humans and pigs [107]. Therefore, the use of hCD31 to a inhibit neutrophil-mediated xenorejection is highly beneficial for preventing xenotransplantation. CD82 is a member of the tetraspanin family of proteins. The widespread expression of CD82 in neutrophils may constitute a strong barrier for the long-term survival of a transplanted xenograft. CD82 has been reported to contribute to the Galα1,3-Gal-independent adhesion of human neutrophils to pECs [108].

## 4. Toll-Like Receptors

TLRs are key PRRs of the immune system required to initiate an effective innate immune response at an early stage of infection, recognizing PAMPs or danger-associated molecular patterns (DAMPs) [109,110]. TLRs are classified into several groups based on the types of PAMPs they recognize. TLR1, 2, 4, and 6 recognize lipids. The second class of TLRs includes TLR5 and 11, which are recognized protein ligands. TLRs 3, 7, 8, and 9 specialize in viral detection and recognize nucleic acids. TLRs are expressed in various cell types, including macrophages, epithelial cells, and ECs, as well as neutrophils, mast cells, basophils, and eosinophils, and play a significant role in the induction of the innate immune responses and the subsequent development of the adaptive immune responses [111,112]. We reviewed recent studies and highlighted that their activation is involved in major aspects of transplantation.

A serious ischemia/reperfusion (I/R) injury is the main relevant factor in solid organ transplant failure. TLRs are activated by endogenous ligands from damaged/stressed cells in an I/R injury. The activation of TLRs-carrying cells triggers the release of inflammatory cytokines and chemokines as well as the recruitment of macrophages, neutrophils, and T cells, causing a full-blown I/R injury. Additionally, neutrophils are the major leukocytes found in I/R injury and promote the progression of I/R tissue damage.

A TLR4 activation can signal through a MyD88-independent pathway (also known as the TRIF-dependent pathway). In the MyD88-independent pathway, TLR4 may recruit the TRIF-related adaptor molecule (TRAM) and TRIF which activates TBK1, thus allowing the activation of interferon-regulatory factor-3 (IRF3). This pathway may also signal through the signaling of NF-κB through the recruitment of TRAF6 recruitment. The activation of the TRIF-dependent pathway induces IFN-β and increases inflammatory cytokine gene expression, causing damage to xenografts (Figure 3).

TLR2 and TLR4 trigger the same pathway, the MyD88 pathway, resulting in the release of NF-κB from IκB, which subsequently stimulates the expression of pro-inflammatory cytokines and pro-apoptotic molecules within the recruited immune cells [113,114] (Figure 3). Furthermore, the stimulation of TLR2 or TLR4 can induce coagulative dysfunctions [115,116]. It is worthwhile to focus on TLR2 and TLR4 as PRRs mediating DAMP-driven injury because the genetic deletion of either TLR2 or TLR4 protects from an I/R injury in a mouse model, possibly by reducing the production of cytokines and chemokines and thus decreasing the neutrophil infiltration and apoptosis [117,118].

The activation of TLR in porcine islets induced both a pro-inflammatory and procoagulant response and thereby contributed to a xenograft rejection [119]. In addition, studies have shown that immunoglobulin antibodies against islet-like cell cluster (ICC) membrane antigens are significantly reduced in the serum of IL-6-deficient mice after transplantation [120]. In addition, discordant pig-to-baboon heart xenografts show a stronger IL-6 response than congruent rhesus monkey-to-baboon xenografts [119]. Therefore, the TLR-mediated induction of IL-6 in porcine organs may contribute to a xenograft rejection.

The interaction of transmembrane receptor TLR with a high mobility group box 1 (HMGB1) has been the focus of research into the pathogenesis of inflammatory diseases and cell damage [121,122]. HMGB1, a DNA-binding nuclear protein, which is released from cells such as monocytes, macrophages, and dendritic cells through specific secretory pathways or is passively released upon cell death, stimulates innate immune mechanisms exerting atypical alarmin functions [123,124,125]. The binding of HMGB to TLRs (TLR2, TLR4, and TLR9) on macrophages activates NF-κB and interferon regulatory factor (IRF) pathways, inducing the production of pro-inflammatory cytokines (TNFα, IL-1, and IFN-I) and chemokines [126,127]. Given its multiple functions in regulating immunity and inflammation, HMGB1 has become the focus of research on the mechanism of allograft rejection. The presence of HMGB1 is associated with heart, kidney, and lung transplantation rejection and liver injury early after transplantation [128,129,130]. Nevertheless, recently, it has also been demonstrated that HMGB1 plays a significant role in mediating an acute xenograft rejection. In rodent cardiac xenotransplantation, hearts from Sprague Dawley (SD) rats were transplanted heterotopically into adult BALB/c mice, and the recipient mice developed acute vascular rejection within 6 days of the transplantation [131]. Meanwhile, after the transplantation, the plasma HMGB1 levels significantly increased in a time-dependent manner from day 1 to the endpoint. Alternatively, immunohistochemical examination showed a strong expression of HMGB1 in the cytoplasm of both rat myocardial cells and infiltrated mouse immune cells in rejected xenografts at the endpoint. A treatment with anti-HMGB1 antibodies significantly suppressed xenoreactive B cell responses and delayed xenoreactive antibody production after xenotransplantation, prolonging cardiac xenograft survival (approximately from 5 to 9 days) [131].

## 5. Xenotransplantation in Clinical Grade

In the last months, different teams have performed clinical transplantations of porcine organs to humans. It was a major milestone in organ transplantation. At the end of 2021, the first kidney xenotransplantation of brain-dead patients was performed by the Robert Montgomery team from New York University (NYU) [132]. The kidney of a GTKO pig was attached to the upper leg blood vessel for 54 h. The porcine kidney produced urine and showed normal creatinine levels, without signs of hyperacute rejection. It is a limited success because both recipients still retain their kidneys and the donor pigs have only one gene modification. In January 2022, another team from the University of Alabama (UAB) implanted kidneys into a human brain-dead decedent in less than 74 h [133]. They used kidneys which came from pigs with ten genes modified, inactivation of GGTA1, CMAH, B4GALNT2, and GHR, and with the additional expression of human CD46, CD55, TM, EPCR, CD47, and HO1. However, the evaluation of the immune rejection in brain-dead patients is relatively limited because their immune systems are different from those of normal patients within the limited time of the observation.

Further work was done on the first heart xenotransplantation. The University of Maryland Medical Center (UMMC) has received the food and drug administration (FDA) approval to perform xenotransplantation in a patient with heart failure without liver or kidney failure. The group conducted a transplant of a porcine heart into a 57-year-old patient, whose immunosuppression was based on a CD40 blockade [134]. The heart from a gene-edited pig had 10 genetic modifications, the same combination of genetic modifications as that used for the orthotopic kidney transplant in a brain-dead patient (see above). In addition, the treatment team also used new experimental anti-rejection drugs. Unfortunately, although the transplanted heart was pulsating well 3 days after surgery, the patient died 2 months later.

Xenotransplantation has not yet been fully developed clinically, mainly because xenoreactive antibodies remain the cause of graft failure [135]. Several efforts have been made to eliminate xenoreactive antibodies on the xenotransplantation, including the development of αGalT KO pigs to eliminate the xenoantigen, to which 70-90% of xenoreactive antibodies are bound and avoid hyperacute rejection. Alternatively, the use of immunosuppressants can improve the success rate of xenotransplantation. However, conventional chemoimmunotherapy being dominated by tacrolimus also has proven insufficient in pig organ xenotransplantation [136]. The kidneys from GalT-KO pigs transplanted into baboons with immunosuppressive agents such as tacrolimus, mycophenolate, and steroids, were rejected by non-α-gal xenoreactive antibodies within 6–16 days [137].

In addition, anti-CD154 mAb is one of the main therapies used in cardiac or renal pig to NHP transplantation and helps to significantly prolong solid-organ survival [37,138,139]. However, because anti-CD154 mAb can directly activate platelets and is associated with thromboembolic complications, the development of this drug change was discontinued [140,141]. Researchers have had to explore other treatment options, but the common protocols currently used for induction and maintenance therapy in solid organ transplants such as the heart, kidney, and pancreas have not worked well in xenografts. Anti-CD40L has been reported to be more effective in preventing graft rejection compared to anti-CD154 therapy in a pig to rhesus macaque xenograft transplant model [142]. The use of anti-CD40L antibodies lacking the crystalline fragment (Fc) effector effect is safe and does not activate the platelets or cause thromboembolism [143,144]. Although the targeted genetic manipulation of donor animals can significantly reduce the immunosuppression, an effective chemical immunization (for example, the application of anti-CD40 Ab, use of anti-thymocyte globulin and rituximab or continuous use of multiple immunosuppressive drugs) is still needed to overcome the antigenic differences between species. Table 2 shows some of the immunosuppressive drugs applied to the xenotransplantation area and their mechanisms of action.

Furthermore, a tolerance induction by mechanisms such as mixed chimerism and thymus transplantation has been explored as a way to overcome the xenogeneic immune barrier.A tolerance induction with combined renal and hematopoietic cell transplantation has been used in the clinical field. Researchers from Stanford University injected donor hematopoietic stem cells with a fixed number of donor T cells after the kidney transplantation [160]. This has proven to be effective in inducing tolerance in an HLA-matched kidney transplantation, allowing for the complete immunosuppressive drug withdrawal in 80% of the study participants. The thymus transplantation strategy is a combination of thymus and organ transplantation from the same donor, resulting in a T-cell tolerance to xenografts and self. Xenogeneic swine thymic transplants induced a tolerance to swine antigens in mice and supported the growth of swine grafts [161]. Another study also showed that in a pig-to-mouse model, murine recipients of fetal pig thymus/liver (FP THY/LIV) grafts achieved an efficient repopulation of CD4+ T cells and maximal pig skin graft tolerance [162]. Additionally, this approach with a powerful tolerance-inducing capacity supported the survival of pig-to-baboon renal xenografts for up to 83 days [37].

The use of a xenoantigen reduction method without the use of transgenes that interfere with the ability of the recipient’s immune system to interact with the xenograft preserves protective immunity within the xenograft intact, allowing the recipient’s immune system to be protected from harmful antibodies in the event of an infection. Taken together, since xenoreactive antibodies are still the cause of xenograft failure, only recipients with negative crossmatches are considered for an initial transplantation in xenografts, which is essential to achieve an extended xenograft survival in human patients.

## 6. Discussion and Future Perspectives

Knowledge of the role of innate immunity in xenograft rejection is rapidly evolving. In most organs, an I/R injury is initiated by the activation of natural antibodies and complements. Moreover, a cellular xenogeneic rejection has also become an important immune barrier to overcome in the clinical use of xenogeneic organs. It is emphasized that innate immunity by macrophages, NK cells, and neutrophils causes severe rejections during xenotransplantation.

Activated innate immune cells destroy allografts not only through direct cytotoxicity but also by promoting the initiation of T cell responses against the graft. The rejection of allografts by the innate immune system involves multiple effector mechanisms that cannot be prevented by a single approach. Our understanding of the mechanism of action of innate immunity to xenoantigen helps to identify the molecules used for the genetic manipulation in donor pigs to suppress the innate alloimmune responses. For example, adding complement regulatory proteins, along with the regulation of coagulation factors, appears to be the basis for a successful xenotransplantation. Furthermore, the transgenic expression of HLA molecules and hCD47 was used to suppress a xenograft rejection by NK cells and macrophages, respectively, and prolong the survival time of the xenograft organs. However, simply the inhibition of antibody-and receptor-mediated xenograft immune responses in animals may not be sufficient without considering the other factors.

In addition, innate immune cells are required for an acute and chronic rejection in certain animal transplant models and by extension perhaps in clinical transplantation. Therefore, the development of strategies for suppressing innate immune cells has considerable potential for the practical applications of xenotransplantation. Activated innate immune cells destroy grafts not only through a direct cytotoxicity but also by promoting the initiation of T cell responses against the graft. Consequently, the tight control of innate immunity such as macrophages, to control the movement of lymphocytes, will be important as the next step in the era of xenotransplantation is realized.

HMGB1 has since been revaluated as a classic DAMP protein that activates various graft cells, leading to dysregulated inflammatory responses and macrophage depletion [163,164]. Recent discoveries have reported a close association between HMGB1 and xenotransplantation. HMGB1 evokes inflammatory responses and accelerates graft rejection. In the long term, we have to think that blocking and managing the HMGB1 signaling axis may increase the transplant’s success and improve the long-term patient outcomes for organ recipients.

So far, this genetic engineering selectively addresses only major interspecific molecular differences, such as eliminating highly immunogenic epitopes, modulating the complement expression, or blocking major immune cell activation pathways. It is worth noting that an interspecific blastocyst complementation and the in vivo generation of organs derived from xenogeneic donor pluripotent stem cells have potential [165]. The Nakauchi team described the possibility of reconstructing xenogeneic organs. Rat pluripotent stem cells (PSCs) were injected into the embryos of mice with an impaired pancreatic development (pdx1 gene knockout) to compensate for embryonic chimerism. The offspring of mice grew a pancreas which derived from the rat cells. The organs functioned normally. Additionally, it can completely compensate for the function of the pancreas in mice with pancreas deficiency [165]. In addition to the pancreas, people have also tried to produce other organs in the body through a blastocyst complementation. Nakauchi et al. complemented the ESCs of rats or mice to the blastocysts of defective mice and grew kidneys from the ESCs of rats or mice in chimeras [166]. In the future, obtaining regenerative organs through complementary blastocysts may be an important approach to avoid the problem of an immune rejection. Still, before xenotransplantation can enter clinical use, more knowledge is needed to fully address the various issues associated with the use of xenotransplantation for human disease. In addition, although xenotransplantation has great application prospects, in the late 1990s, regulators raised concerns about a cross-infection with heterogeneous pathogens. The safety and ethical issues associated with xenotransplantation remain and need to be studied in depth.

## Figures and Tables

**Figure 1 cells-11-03865-f001:**
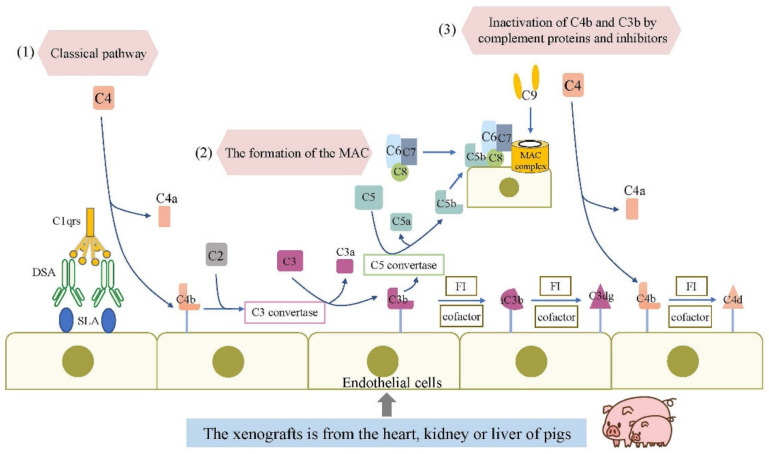
Complement activation in antibody-mediated xenorejection. (1) The classical pathway of complement is activated when antibodies (referred to as DSA) bind to SLAs expressed on endothelial cells of the transplanted organ. Classical pathway activation causes the cleavage of C4, and C4b is covalently attached to target surfaces. C4b can go on to form convertases and soluble mediators that perpetuate and amplify the complement cascade. C3b can covalently connect with target cells and regulate them for enhanced phagocytosis. (2) Activation of C3 leads to the formation of a C5 convertase, which cleaves C5 into C5a and C5b. C5b recruits several other complement proteins to the cell surface and ultimately initiates the formation of the MAC. MAC forms pores in the cell membrane and causes cell lysis at high concentrations. (3) The C4b and C3b molecules can be targeted by complement regulatory proteins and inhibitors such as FI to inactivate them. Although they are no longer catalytically active, the C4d and C3dg fragments remain covalently linked to the target cells and are a marker of early complement activation. Abbreviations: DSA, donor-specific antibodies; MAC, membrane attack complex; SLA, swine leukocyte antigen; FI, factor I.

**Figure 2 cells-11-03865-f002:**
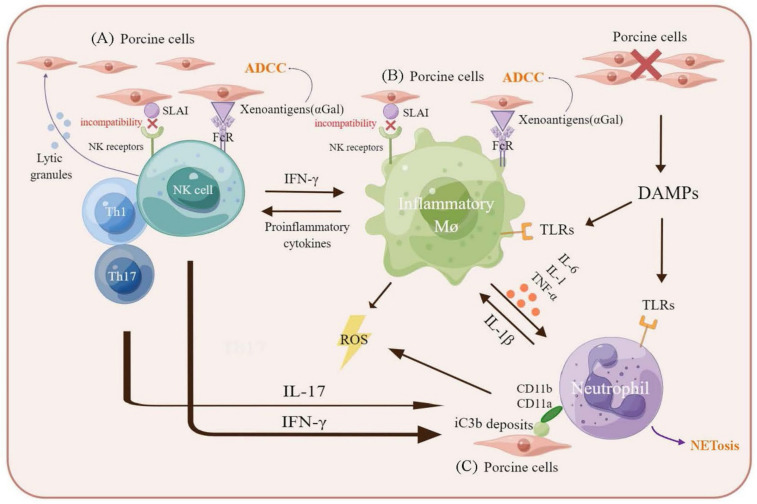
Schematic representation of Innate immune cellular-mediated rejection in xenograft. (A) NK cells-mediated rejection. The Fc portion of the xenogeneic cell antibody is recognized by FcRs of NK cells, releasing of lytic granules (marked with blue spots) from NK cells, promoting cell injury. SLA class I does not bind to human inhibitory NK receptors to trigger NK cell-mediated xenogeneic cell lysis. (B) Macrophages-mediated rejection. Macrophages can be activated by IFN-γ that are produced by xenoreactive T cells. Activated macrophages contribute to cellular xenograft rejection by amplifying T cell-mediated immune responses. Macrophages can also be activated by the FcR and TLRs-mediated signalling in interaction with xenoreactive-antibody-coated cells. Macrophages secrete proinflammatory cytokines that augmented the cytotoxicity of macrophages and activate neutrophils to promote NETosis. Especially, IL-8 contributes to neutrophil recruitment and IL-1β enhances NETosis formation in neutrophils. Activated NK cells and macrophages induce ADCC against cells. (C) Moreover, the neutrophil extracellular traps of neutrophils can also activate macrophages as DAMPs. Macrophages activate NK cells and IFN-γ from activated NK cells can activate neutrophils. CD11a and CD11b on neutrophils combine with iC3b deposits on porcine cells, suggesting that neutrophils can recognize porcine target cells via binding of CD11a, CD11b and iC3b. The figure is drawn by Figdraw. Abbreviation: NK, natural killer; Mø, macrophage; FcR, Fc receptor; SLA, swine leukocyte antigen; DAMPs, damage-associated molecular patterns; TLRs, Toll-like receptors; IFN, interferon; TNF, tumor necrosis factor; IL, interleukin; ADCC, antibody-dependent cellular cytotoxicity.

**Figure 3 cells-11-03865-f003:**
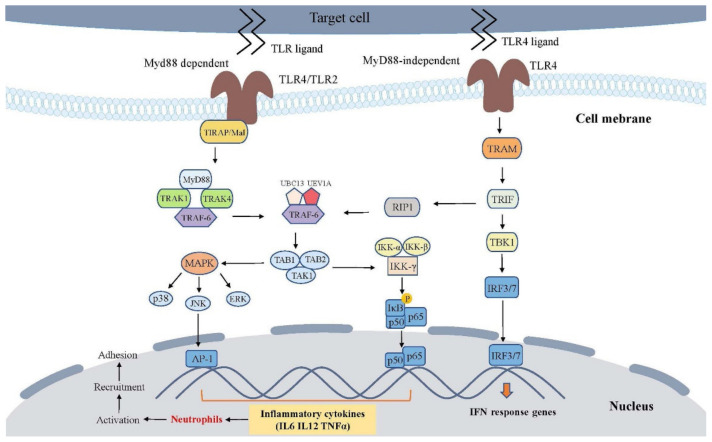
TLRs signaling overview. This figure depicts the endogenous ligands released during xenotransplantation that trigger two of the most important TLR receptors, TLR2 and TLR4, and the MyD88-dependent and MyD88-independent (TRIF-dependent) signaling pathways activated by TLRs. Abbreviation: TLR, toll-like receptor; MyD88, myeloid differentiation primary response protein 88; Mal, MyD88-adapter like; TIRAP, TIR domain-containing adapter; TRAK, TSH-receptor antibody test; TRAM, Trif-related adaptor molecule; TRIF, TIR domain-containing adaptor protein inducing type 1 interferons; TBK1, TANK binding Kinase-1; IRF, interferon regulatory factor; MAPK, mitogen-activated protein kinases; JNK, c-Jun N-terminal kinase; ERK, extracellular signal-regulated kinase; RIP1, Receptor-Interacting Protein 1; TRAF6, tumor necrosis factor-receptor-associated factor-6; TAK1, TGF-β-activated kinase-1; TAB1, TAK1-Binding Protein-1; TAB2, TAK1-Binding Protein-2; UBC13, ubiquitin-conjugating enzyme-13; UEV1A, ubiquitin-conjugating enzyme E2-variant-1; IKK, IkappaB kinase; AP-1, transcription factor complex-1; interleukin, IL; IFN, interferon; TNF, tumor necrosis factor.

**Table 1 cells-11-03865-t001:** Components of innate immunity that recognize a Xenograft.

Immune Response	Component	Rejection Mechanism	Target Gene Editing	Rejection Period
Innate	Complement alternative pathway	Inhibits complement activation	factor H	HAR
	Complement regulatory proteins	Complement regulation	hCD55; hCD46; hCD59 expression	HAR; AVR
	Natural antibody	Reduces natural anti- αGal and non-Gal antibodies responses	GGTA1-KO; CMAH-KO; β4GALNT2-KO	HAR; AVR
	Platelet/Thrombin	Coagulation/thrombosis reduction	Human Thrombomodulin; human CD39 expression	AVR
	Macrophage	Macrophages regulation	hCD47 expression (a marker for “self”); human signal regulatory protein alpha expression; human heme oxygenase 1 expression; human A20 expression; human CD39 expression.	AVR
	NK cell	Natural NK cells regulation	human HLA-G, HLA-E expression; human beta2 microglobulin expression	AVR
	Neutrophil	Neutrophils regulation	Integrins	AVR

Abbreviations: αGal, galactose alpha-1,3-galactose; αGalT-KO, alpha1,3-galactosyltransferase-knockout; MHC: major histocompatibility complex; NK, natural killer; HLA, human leukocyte antigen; hCD47, human CD47; HAR, hyperacute rejection; AVR, acute vascular rejection.

**Table 2 cells-11-03865-t002:** Immunomodulating agents applied to xenotransplantation.

Sources	Immunomodulating Agents	Component Mechanism	Pig-to-NHP Xenotransplantation	Reference
Under investigation	Anti-CD154 Ab	Blockade of CD40−CD154	Kidney, heart, islets	[145,146,147]
	Cobra venom factor	Inhibition of complement system	Islets	[34]
	Anti-CD40 Ab	Blockade of CD40−CD154 costimulatory signal	Kidney, heart, islets	[145,147,148]
FDA non-approved, commercially available	Eculizumab	Blockade of the C5b-9 MAC	Pancreas	[149]
	Etanercept	Anti-TNF-alpha inhibitor	Islets	[150]
	Intravenous immunoglobulin	Modulating antigen presenting cell activity and complement activation	Hearts	[151,152]
	IL-1 receptor antagonist	IL-1 inhibitor	Heart	[153]
	IL-6 receptor antagonist	IL-6 inhibitor	Islet, Kidney, Heart	[154,155]
FDA approved for transplantation	Tacrolimus	Inhibition of the enzyme calcineurin by binding of FKBP-12	Pancreas	[156,157]
	Mycophenolic acids	Inhibition of the lymphocyte cycle by blocking inosine monophosphate dehydrogenase	Kidney, heart, islets	[158,159]

Abbreviation: Ab, antibody; NHP, non-human primate; MAC, membrane attack complex; TNF, tumor necrosis factor; IL, interleukin; FDA, the Food and Drug Administration.

## Data Availability

Not applicable.

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
