# Peer review of "Advances in Innate Immunity to Overcome Immune Rejection during Xenotransplantation"

_cells, 2022, doi:10.3390/cells11233865_

Round 1

Reviewer 1 Report

A brief summary:

This review addresses the current known innate immunity barriers for xenotransplantation, including both innate humoral and cellular xenoimmune responses in xenograft rejection as well as the strategies such as genetic modifications of source pigs and immunosuppressive therapy to overcome these hurdles. This article provides an overview of the key components and actions of the innate immune system that are involved in rejection of xenografts in pig-to-primate (human and non-human) xenotransplantation, and highlights progress achieved through gene editing technologies, and immunosuppression treatments to overcome barriers and key issues for future investigations to successful clinical xenotransplantation.

General concept comments:

1. With the advances in immunosuppressive and tolerance induction protocols and genetic modifications of porcine source animals, the survival of xenografts in recipients has greatly extended to months, and thus the chronic rejection would occur. There are several similar review articles have been published on innate immunity in acute xenorejection, but however the roles of innate immunity in chronic xenorejection have not be clearly reviewed. By including the chronic rejection caused by innate immunity, this review article would be more relevant and of interest to the scientific community by identifying the gap in knowledge. On page 14, the article mentioned that “In addition, innate immune cells are required for acute and chronic rejection in certain animal transplant models and by extension perhaps in clinical transplantation”. How innate immunity is involved in chronic rejection could be further discussed. Then, this review article would be more comprehensive if it covers the chronic rejection topic.

2. The tolerance induction strategies such as mixed chimerism and thymic transplantation to overcome innate immune xenogeneic reactions could be reviewed since they could also be used in the combination therapy to successful clinical xenotransplantation.

Specific comments:

3. In Abstract section, in this sentence “Recently, heart and kidney from pig were transplanted into patients,”. For kidney transplantation, “the patients” is suggested to be replaced with recipients as the cited papers used because kidneys were transplanted to human brain-dead decedents in recent two xenotransplantation experiments. Therefore, for kidney xenotransplantation, using “recipients” would be more appropriate than “patients”.

4. In 2. The complement and natural antibodies system section, the references related to the studies using Triple-Knockout (TKO) pigs [e.g., Martens, G. R. et al. Transplantation 101, e86–e92 (2017), Yamamoto, T. et al. Sci. Rep. 10, 9771 (2020), and Ma, D. et al. Am. J. Transplant. 22, 46–57 (2022)] are suggested to be cited and discussed in this section.  Hence, on page 5, the statement “At present, we hold that deletion of three identified xenoantigens with the expression of one or more hCRPs would be the foundation for future modification in pigs.” could be rephrased to update the current development of genetically engineered pigs.

5. On page 5, the sentence “At the same time, the study has shown that by supplementing primate blood coagulation components, massive bleeding of receptors can be avoided” is not clear.

6. On page 6, the text “Genetic induction of hCD47 on porcine cells reduces the susceptibility of porcine cells to phagocytosis by human macrophages in vitro and prevents human HCs from rejection by macrophages in mice carrying an allele of SIRPα that productively binds hCD47” is confusing that why human HCs are mentioned.

7. On page 7, the statement “Therefore, transgenic expression of α-2,6-sialyl transferase (α 2,6-ST) induces the binding of the α-2,6-sialylated ligands to CD33-related Siglecs on macrophages, thereby inhibiting macrophage-mediated xenocytotoxicity in vitro. The findings suggested that α 2,6-ST transgenic expression may contribute to macrophage-mediated xenocytotoxicity.” has not listed the supported citations.

8. On page 12, the text “implanted kidneys into a human brain-dead decedent” not “implanted kidneys into brain-dead patients orthotopic” should be written for precise description according to the cited reference.

9. On page 12, the sentence “Baboons immunosuppressed with tacrolimus, mycophenolate, and steroids, were rejected by non-α-gal xenoreactive antibodies within 6–16 days.” misses to put kidneys.

10. On page 12, in this sentence “Although targeted genetic manipulation of donor animals can significantly reduce immunosuppression, effective chemical immunization is still needed to overcome antigenic differences between species.”, the “effective chemical immunization” needs to be more clarified.

11. For Figures, it would be more appropriate if the xenografts are also depicted in the figure 1 for their actions to activate complements? The figure 1 caption describes “Human leukocyte antigens (HLA) expressed on endothelial cells of the transplanted organ”. It should be SLAs since xenotransplantation is discussed in this figure. Thus it would be easy for the readers to interpret and understand this figure? So is the figure 3.

Minor comments

12. Several reference brackets should have a space between the last word.  For example: HAR[31,32], grafts[49]; survival[51], etc.

Author Response

A brief summary

This review addresses the current known innate immunity barriers for xenotransplantation, including both innate humoral and cellular xenoimmune responses in xenograft rejection as well as the strategies such as genetic modifications of source pigs and immunosuppressive therapy to overcome these hurdles. This article provides an overview of the key components and actions of the innate immune system that are involved in rejection of xenografts in pig-to-primate (human and non-human) xenotransplantation, and highlights progress achieved through gene editing technologies, and immunosuppression treatments to overcome barriers and key issues for future investigations to successful clinical xenotransplantation.

Response: We sincerely appreciate your constructive comments and suggestions. It is very valuable and helpful to us. We tried our best to amend the manuscript according to the reviewers’ comments, and all suggestions have been incorporated into the revised manuscript. The changes in the revised manuscript with highlight. I believe our paper become more logical and clearer. Of course, we hope to hear reviewer’s comments and continue to promote this paper.

General concept comments

  1. With the advances in immunosuppressive and tolerance induction protocols and genetic modifications of porcine source animals, the survival of xenografts in recipients has greatly extended to months, and thus the chronic rejection would occur. There are several similar review articles have been published on innate immunity in acute xenorejection, but however the roles of innate immunity in chronic xenorejection have not be clearly reviewed. By including the chronic rejection caused by innate immunity, this review article would be more relevant and of interest to the scientific community by identifying the gap in knowledge. On page 14, the article mentioned that “In addition, innate immune cells are required for acute and chronic rejection in certain animal transplant models and by extension perhaps in clinical transplantation”. How innate immunity is involved in chronic rejection could be further discussed. Then, this review article would be more comprehensive if it covers the chronic rejection topic.

Response: Thanks for your suggestion. We have added some information about chronic rejection section. The following part was added: Chronic rejection is also a potential cause of clinical allograft failure. Indeed, xenograft vasculopathy has been reported in the hearts transplanted from hCRP transgenic pigs to primates, which was considered to be a characteristic feature of chronic rejection [23,24]. The main manifestations are the proliferation of endothelial cells in the graft vessels, vessel narrowing, and thrombosis, which eventually leads to graft failure [25]. Fortunately, the use of anti-CD40 mAb on transgenic pig hearts (GTKOhTg.hCD46.hTBM) blocked the same CD40-CD40 ligand costimulation path-way to avoid thrombotic microangiopathy and allow successful survival of pig cardiac xenografts in baboons for 236 days [26]. The result suggested that anti-CD40 mAb can achieve long-term cardiac xenograft survival. And the expression of hCD39 on mouse hearts attenuated platelet deposition and small vessel thrombosis, thereby prolonging graft survival, and cardiac xenografts from CD39-null mice in rats showed reduced survival times [27]. CD4 + T cells may be a cause of chronic rejection. The depletion of CD4 T cells maintained the survival of pig to rhesus macaque kidney xenografts for up to 499 days [28]. (Page 4 in the revised manuscript)

  1. The tolerance induction strategies such as mixed chimerism and thymic transplantation to overcome innate immune xenogeneic reactions could be reviewed since they could also be used in the combination therapy to successful clinical xenotransplantation.

Response: Thanks for your nice comments. We have provided additional information about mixed chimerism and thymic transplantation (Pages 13 and 14 in the revised manuscript).

Specific comments:

  1. In Abstract section, in this sentence “Recently, heart and kidney from pig were transplanted into patients,”. For kidney transplantation, “the patients” is suggested to be replaced with recipients as the cited papers used because kidneys were transplanted to human brain-dead decedents in recent two xenotransplantation experiments. Therefore, for kidney xenotransplantation, using “recipients” would be more appropriate than “patients”.

Response: Thanks for this constructive suggestion. We have revised “patients” to “recipients” in the current manuscript.

  1. In 2. The complement and natural antibodies system section, the references related to the studies using Triple-Knockout (TKO) pigs [e.g., Martens, G. R. et al. Transplantation 101, e86–e92 (2017), Yamamoto, T. et al. Sci. Rep. 10, 9771 (2020), and Ma, D. et al. Am. J. Transplant. 22, 46–57 (2022)] are suggested to be cited and discussed in this section. Hence, on page 5, the statement “At present, we hold that deletion of three identified xenoantigens with the expression of one or more hCRPs would be the foundation for future modification in pigs.” could be rephrased to update the current development of genetically engineered pigs.

Response: Thanks for your good comments. The references you provided are great helpful for us to have a comprehensive understanding of the current development of genetically engineered pigs. We have provided additional information and discussion in this section and cited all the references mentioned (Page 5 and References 48, 49, and 51 in the revised manuscript).

  1. On page 5, the sentence “At the same time, the study has shown that by supplementing primate blood coagulation components, massive bleeding of receptors can be avoided” is not clear.

Response: Sorry for the inaccurate description. We have corrected it to “At the same time, the study has shown that by supplementing primate blood coagulation components, massive hemorrhage in the setting of liver xenotransplantation can be avoided.” (Page 5 in the revised manuscript)

  1. On page 6, the text “Genetic induction of hCD47 on porcine cells reduces the susceptibility of porcine cells to phagocytosis by human macrophages in vitro and prevents human HCs from rejection by macrophages in mice carrying an allele of SIRPα that productively binds hCD47” is confusing that why human HCs are mentioned.

Response: Sorry for the error. We reworded this sentence as suggested (Page 7 in the revised manuscript).

  1. On page 7, the statement “Therefore, transgenic expression of α-2,6-sialyl transferase (α 2,6-ST) induces the binding of the α-2,6-sialylated ligands to CD33-related Siglecs on macrophages, thereby inhibiting macrophage-mediated xenocytotoxicity in vitro. The findings suggested that α 2,6-ST transgenic expression may contribute to macrophage-mediated xenocytotoxicity.” has not listed the supported citations.

Response: Sorry for the mistake. We have listed the supported citations (Reference 78 in the revised manuscript).

  1. On page 12, the text “implanted kidneys into a human brain-dead decedent” not “implanted kidneys into brain-dead patients orthotopic” should be written for precise description according to the cited reference.

Response: Sorry for the confusion. We have already corrected it to “implanted kidneys into a human brain-dead decedent” (Page 12 in the revised manuscript).

  1. On page 12, the sentence “Baboons immunosuppressed with tacrolimus, mycophenolate, and steroids, were rejected by non-α-gal xenoreactive antibodies within 6–16 days.” misses to put kidneys.

Response: Sorry for the inaccurate description. We have it corrected as suggested (Page 13 in the revised manuscript).

  1. On page 12, in this sentence “Although targeted genetic manipulation of donor animals can significantly reduce immunosuppression, effective chemical immunization is still needed to overcome antigenic differences between species.”, the “effective chemical immunization” needs to be more clarified.

Response: Sorry for the confusion. We have added some information about “effective chemical immunization”. For example, the application of anti-CD40 Ab, the use of anti-thymocyte globulin and rituximab, or the continuous use of multiple immunosuppressive drugs (Page 13 in the revised manuscript).

  1. For Figures, it would be more appropriate if the xenografts are also depicted in the figure 1 for their actions to activate complements? The figure 1 caption describes “Human leukocyte antigens (HLA) expressed on endothelial cells of the transplanted organ”. It should be SLAs since xenotransplantation is discussed in this figure. Thus, it would be easy for the readers to interpret and understand this figure? So is the figure 3.

Response: Thanks for your kind suggestion. In the revised manuscript, we have corrected these mistakes as suggested.

Minor comments

  1. Several reference brackets should have a space between the last word. For example: HAR[31,32], grafts[49]; survival[51], etc.

Response: Thank you for the reminders, we have added a space between the last word and the reference.

Reviewer 2 Report

Nice review. The only weakness is the chapter on clinical application, p12:

In the Maryland case, anti-CD40ab was used from Kiniksa.

Tonix has now a new anti-CD40L, which seems to be non-thrombogenic. (T Kawai, 2000 on renal XTX versus 2022 again on renal XTX)

Author Response

    Thanks for your positive comments and valuable suggestion. We have added that the immunosuppression used in the Maryland case was anti-CD40ab. And we have added some information “Anti-CD40L has been reported to be more effective in preventing graft rejection compared to anti-CD154 therapy in a pig to rhesus macaque xenograft transplant model [142]. The use of anti-CD40L antibodies lacking the crystalline fragment (Fc) effector effect is safe and does not activate platelets or cause thromboembolism.”

In addition, we amended our manuscript in the re-submitted word, including providing more discussions about current reports, updating the latest progress, modifying figures, as well as correcting some other spelling mistakes. The changes with highlight in the revised manuscript. We hope that the recent manuscript changes make the manuscript more readable and more acceptable for publication.

Round 2

Reviewer 1 Report

The authors have responded well to the comments. 

However, in Figure 1, HLA is still marked on the figure and appears in the caption.

Author Response

Sorry for the error. We have corrected “HLA” to “SLA” in the Figure 1 and in the caption.